# Delivery of siRNA to Ewing Sarcoma Tumor Xenografted on Mice, Using Hydrogenated Detonation Nanodiamonds: Treatment Efficacy and Tissue Distribution

**DOI:** 10.3390/nano10030553

**Published:** 2020-03-19

**Authors:** Sandra Claveau, Émilie Nehlig, Sébastien Garcia-Argote, Sophie Feuillastre, Grégory Pieters, Hugues A. Girard, Jean-Charles Arnault, François Treussart, Jean-Rémi Bertrand

**Affiliations:** 1LuMIn, CNRS, ENS Paris-Saclay, CentraleSupélec, Université Paris-Saclay, 91405 Orsay, France; sandra.claveau@live.fr; 2Vectorologie et Thérapeutiques Anticancéreuses, CNRS, Institut Gustave Roussy, Université Paris-Saclay, 94805 Villejuif, France; jean-remi.bertrand@gustaveroussy.fr; 3SCBM, Institut Joliot, CEA, Université Paris-Saclay, 91191 Gif-sur-Yvette, Francesebastien.garcia-argote@cea.fr (S.G.-A.); sophie.feuillastre@cea.fr (S.F.); gregory.pieters@cea.fr (G.P.); 4Diamond Sensors Laboratory, Institut LIST, CEA, Université Paris-Saclay, 91191 Gif-sur-Yvette, France; hugues.girard@cea.fr (H.A.G.); jean-charles.arnault@cea.fr (J.-C.A.); 5Institut d’Alembert, CNRS, ENS Paris-Saclay, Université Paris-Saclay, 91190 Gif-sur-Yvette, France

**Keywords:** nanodiamond, tritium, biodistribution, Ewing sarcoma, drug delivery, siRNA, nanomedicine

## Abstract

Nanodiamonds of detonation origin are promising delivery agents of anti-cancer therapeutic compounds in a whole organism like mouse, owing to their versatile surface chemistry and ultra-small 5 nm average primary size compatible with natural elimination routes. However, to date, little is known about tissue distribution, elimination pathways and efficacy of nanodiamonds-based therapy in mice. In this report, we studied the capacity of cationic hydrogenated detonation nanodiamonds to carry active small interfering RNA (siRNA) in a mice model of Ewing sarcoma, a bone cancer of young adults due in the vast majority to the *EWS-FLI1* junction oncogene. Replacing hydrogen gas by its radioactive analog tritium gas led to the formation of labeled nanodiamonds and allowed us to investigate their distribution throughout mouse organs and their excretion in urine and feces. We also demonstrated that siRNA directed against *EWS-FLI1* inhibited this oncogene expression in tumor xenografted on mice. This work is a significant step to establish cationic hydrogenated detonation nanodiamond as an effective agent for in vivo delivery of active siRNA.

## 1. Introduction

The use of nanoparticles as vectors for drug delivery has been largely described by the scientific community during these past decades, with numerous applications [1,2]. Nanoparticles facilitate intracellular delivery and protection of the cargo against degradation, therefore they present many advantages for the vectorization of small nucleic acids such as small interfering RNA (siRNA). The latter are used to control gene expression by silencing targeted genes. Considering their intrinsic poor cellular penetration and low stability in biologicals medium [3,4], siRNA must be associated to an effective carrier. Different types of siRNA transporting agents have been reported either organic (e.g., liposomes, cationic polymers or dendrimers [5]), inorganic (e.g., metallic nanoparticles such as gold, iron, titanium [4]) or mineral like clay [6], silica nanoparticles [7] and nanodiamonds [8,9]. Most inorganic nanocarriers present a good-to-high chemical stability; they have a low toxicity at therapeutic dose of the drug:carrier conjugate and are able to deliver their cargo compounds into cells. Nevertheless, the high stability of inorganic nanoparticles goes with the fact that they are not biodegradable, and for pharmacological applications, this can be a limiting factor. Therefore, for the safe development of these particles, it is crucial to determine their possible elimination pathways after administration and study how they are distributed in the body.

Here, we report on the use of cationic detonation nanodiamonds (DND) for the delivery of siRNA directed against Ewing sarcoma (ES) junction oncogene *EWS-FLI1* to ES tumor xenografted on mice. Ewing sarcoma is a rare bone and soft tissue cancer [10] which is caused in the vast majority of cases by the formation of *EWS-FLI1* oncogene. To carry siRNA, DND surface needs to be modified to be able to bind negatively charged nucleic acids by electrostatic interactions. One strategy relies on cationic polymer coating of diamond surface, which is either done by electrostatic interaction [9,11] or by covalent grafting [12,13]. However, polymer coating may lead to the formation of large aggregates by crosslinking DND to other DND via polymer chains bridging. Another strategy, the one selected for this study, is based on the direct surface modification of DND using hydrogen gas combined with microwave plasma or with annealing method. Both hydrogenation methods were recently compared [14]. It was shown that after such reductive treatment, the hydrogenated DND (H-DND) can be dispersed in water, and acquire a positive surface charge characterized by a zeta potential of ≈+50 mV [15]. In a previous work, we described that H-DND can carry efficiently siRNA to ES cultured cells and promote specific targeted *EWS-FLI1* oncogene inhibition [16]. We want to extend this strategy to preclinical study now. Considering the very high chemical stability of diamond, these further investigations have to consider the risk of accumulation in the body after inoculation, as already described for larger nanodiamonds (around 50 nm diameter), produced by a different process than detonation [17]. However, owing to their small unitary size (3–8 nm) DND are good candidates for in vivo applications. Indeed, particles smaller than the filtration cutoff of kidney can be eliminated through urines, after glomerular passage, and since this limit for kidney is around 7–10 nm (depending of the molecule shape) [18,19], H-DND could be eliminated through this pathway. Indeed, Riojas et al. [20] showed that hydroxylated 7-nm sized DND further functionalized with radiolabeled amino groups are efficiently eliminated in urines if the solution is filtered before being intravenous injected.

In this work, we describe an original method that we developed to treat ES in vivo by injecting DND:siRNA complexes, and to determine the organ distribution of these DND. To this aim, we used DND labeled with tritium by annealing method [21]. We show that siRNA (i) can be loaded onto hydrogenated or tritiated DND (H-DND and T-DND, respectively), (ii) can efficiently inhibit *EWS-FLI1* in ES tumor model xenografted on mice and (iii) that the organ distribution and elimination of T-DND can be measured thanks to its radioactivity.

## 2. Materials and Methods

### 2.1. Preparation of Hydrogenated and Tritiated DND and Associated Suspensions

The procedures are detailed in reference [21]. DND (Reference: G02 grade, primary size 3–8 nm, from PlasmaChem GmbH, Berlin, Germany) were first manually milled in a mortar and then annealed using appropriate gas.

#### 2.1.1. H-DND Aqueous Suspension

A mass of 30 mg of as-received and manually milled DND were placed in a quartz tube (3.5 mL) with an isolation valve, an in/out gas connection and connected to a cold trap. Vacuum was created and H_2_ gas was loaded at 250 mbar pressure. The tube was placed in an oven, and connection was made with a trapping set-up. The oven was turned on during 1 h at 550 °C. The powder was then pumped for 30 min before disconnection and air exposure. Particles were dispersed in ultrapure water (resistivity: 18.2 MΩ.cm) and sonicated (Model UP400S, 300 W and 24 kHz, from Hielscher GmbH, Teltow, Germany) for 1 h under a cooling system. Aggregates of particles were then removed by centrifuging the suspension for 40 min (acceleration: 2400× *g*, speed: 4754 rpm) and the supernatant was collected, forming the stock H-DND aqueous suspension. The final concentration was calculated by measuring the mass of particles after drying a calibrated volume of the supernatant.

#### 2.1.2. Mixture of T-DND and H-DND (Later Denoted T-DND to Simplify) Aqueous Suspension

As-received manually milled DND were first pre-oxidized to remove native C-H groups to achieve a more quantitative control of tritium added to the surface [21]. To this aim, 100 mg was placed in an alumina crucible under air for 1 h 30 min at 550 °C. Then, 30 mg of these pre-oxidized DND were annealed for 4 h at 550 °C, using the same process that for H-DND with 10% T_2_ and 90% H_2_ gas mixture (in order to obtain the desired activity). To remove all the labile tritium atoms, the treated powder was poured into methanol (4 mL) and the solvent was evaporated. The operation was repeated twice and the powder was then stored dried under a nitrogen atmosphere. Quantification of the tritium incorporation was assessed by measuring the activity (using liquid scintillation counting) in the combustion gas after the total combustion of T-DND in air (3 h at 600 °C). We measured a total specific activity of 13 mCi.mg^−1^.

The synthesized T-DND (4 mg) were dispersed in ultrapure water (3 mL) and sonicated (3 mm conical microprobe Vibracell 75043, 750 W, 28% amplitude, Sonics & Materials, Inc., Newton, CT, USA) for 1 h under a cooling system. To remove highly aggregated particles, the suspensions were centrifuged for 40 min (2400× *g*, 4754 rpm) followed by supernatant separation. A liquid scintillation counting was performed on the supernatant, yielding a specific activity of 18 mCi/mL.

To reduce the high specific activity of the suspension, T-DND from the supernatant (10 µL) were mixed with a solution of H-DND (2 mg/mL, 2.1 mL). The later was prepared following Section 2.1.1 method with a small adjustment: in order to be consistent with the tritium treatment, this H-DND suspension was also prepared from pre-oxidized DND treated for 4 h under H_2_ at 550 °C. The final DND solution was split in 3 sealed vials (0.7 mL each). The total activity measured by liquid scintillation counting of this final solution was 97 µCi/mL.

#### 2.1.3. Additional Pre-Injection Washing of T-DND Solution.

Before injection to mice, the T-DND solution was centrifugated (acceleration 10,600× *g* for 3 h at 10 °C) in order to eliminate residual labile tritium atoms by exchange with water. Washed T-DND could then be stably suspended in water, and only 2% of the total initial tritium radioactivity was lost in the supernatant (see Appendix A).

### 2.2. Hydrodynamic Size and Electrophoretic Mobility Characterizations DND Solution

Hydrodynamic size and zeta potential of H-DND and DND:siRNA complexes in solution were measured by dynamic light scattering (DLS) using a NanoBrook 90Plus PALS (Brookhaven Instruments, Holtsville, NY, USA) in 1 cm thick cuvette in deionized water. Hydrodynamic sizes are inferred from the scattered intensity autocorrelation function. The latter was then analyzed with the non-negative constrained least squared method [22], which is one of the methods of reference to infer the size from polydisperse suspensions. The size values reported correspond to the dominant population. For radioactive nanodiamonds, measurements were performed in a sealed cuvette.

### 2.3. siRNA Sequences and Binding siRNA to Hydrogenated or Tritiated DND and siRNA Binding Capacity Assay

#### 2.3.1. siRNA Sequences

siRNA was purchased from Kaneka Eurogentec S.A. (Seraing, Belgium). The sequence complementary to the *EWS/FLI1* fusion oncogene (siAS) is: sense strands 5′-GCAGCAGAACCCUUCUUAUd(GA)-3′ and antisense strand 5′-AUAAGAAGGGUUCUGCUGCd(CC)-3′. The control irrelevant sequence (siCt) is: sense strand 5′-CGUUACCAUCGAGGAUCCAd(AA)-3′ and the antisense strand 5′-UGGAUCCUCGAUGGUAACGd(CT)-3′.

#### 2.3.2. Binding siRNA to Hydrogenated or Tritiated DND and siRNA Binding Capacity Assay

siRNA complexation to cationic DND was performed by slowly dropping a siRNA solution to cationic DND solutions placed in a sonication bath (Ultrasonic cleaner, VWR International S.A.S., Fontenay-sous-Bois, France) at its maximum power during 10 min, maintaining room temperature in the bath. The measurement of hydrodynamic size and zeta potential were performed afterwards. The determination of the binding capacity of siRNA to nanodiamonds was performed by mixing to a fixed concentration of siRNA (0.3 µg/mL) an increasing concentration of H-DND from 0 to 600 µg/mL in 100 mM NaCl, 10 mM HEPES pH 7.2 buffer, in a fixed 60 µL final volume. After centrifugation (16,000× *g* at 10 °C for 15 min), non-complexed free siRNA concentrations were measured on 30 µL of the supernatants to which an equal volume of 1 µg/mL ethidium bromide (EtBr, Sigma-Aldrich S.a.r.l, Saint-Quentin Fallavier, France) was added. The mixtures were placed into a 96-wells plate, then analyzed with a fluorescence plate reader (Glomax Multi+, Promega, Charbonnières-les-Bains, France) set at 525 nm excitation and 580–640 nm bandpass detection wavelengths, to infer the free siRNA amount. The results are presented as the fraction of the fluorescence intensity relative to the one of the total amounts of siRNA before adding DND. Experiments were realized in triplicate.

### 2.4. Measurement of EWS-FLI1 Inhibition in Cultured Cells by RT-qPCR

One day before treatment, 3 × 10^5^ human Ewing sarcoma cells A673 were seeded in 12 wells-plate in DMEM medium (Gibco, Life Technologies S.A.S., Courtaboeuf, France) containing 10% fetal calf serum (Gibco, Life Technologies S.A.S., Courtaboeuf, France) and 1% Penicillin, streptomycin solution (Gibco, Life Technologies S.A.S., Courtaboeuf, France) and then incubated at 37 °C, with 5% CO_2_ and 95% hygrometry. Then the medium was discarded and replaced by 500 µL of 75 nM siRNA bound to H-DND at a mass ratio of 5:1, 25:1 or 50:1 (H-DND:siRNA) in DMEM medium containing 10% fetal calf serum and 1% penicillin and streptomycin solution, for 24 h at 37 °C, 5% CO_2_ and 95% hygrometry. Comparatively, same conditions are used with 75 nM of siAS bounded to Lipofectamine 2000 (Life Technologies S.A.S., Courtaboeuf, France) added to cells in serum containing medium. Then, the medium was discarded, and the cells were lysed by 400 µL of Trizol solution (Invitrogen) and collected in Eppendorf tubes. Total RNA were extracted by adding 60 µL of chloroform:isoamyl alcohol (49:1). After centrifugation at 13,000 rpm for 15 min at 10 °C, 150 µL of the aqueous phase was precipitated by adding 150 µL isopropanol for 15 min at room temperature followed by centrifugation for 15 min at 13,000 rpm. Pellets were washed twice with 70% ethanol solution and dried. Plellets were then solubilized with water (10 µL) containing 0.5 U/µL of RNasin (Promega, Charbonnières-les-Bains, France) and RNA was quantified with a spectrophotometer (NanoDrop™, Life Technologies S.A.S., Courtaboeuf, France). The cDNA was prepared by heating 1.5 µg of RNA in 12.5 µL of water at 75 °C for 5 min and with 2 µL of 50 µg/mL random primer (Promega). Then, 4 µL of 5X M-MLV buffer (Promega), 0.5 µL dNTP 20 mM (Promega), 0.5 µL RNasin 40 U/µL (Promega) and 0.5 µL M-MLV Reverse Transcriptase 200 U/µL (Promega) were added and incubation was performed for 15 min at 25 °C followed by 1 h at 42 °C. *EWS-FLI1* mRNA expression is performed by qPCR on 5 µL of 1/20 diluted cDNA, 0.4 µL of each primer at 10 mM concentration, 4.2 µL deionized water and 10 µL of 2X KAPA SYBR^®^ FAST Master Mix (Sigma-Aldrich S.a.r.l, Saint-Quentin Fallavier, France). PCR was performed for 40 cycles in fast mode on a StepOnePlus™ system (Applied Biosystems, Life Technologies S.A.S., Courtaboeuf, France). *EWS-FLI1* gene was amplified with *EWS*-forward primer: 5′-AGC AGT TAC TCT CAG CAG AAC ACC-3′ and *FLI1*-reverse: 5′-CCA GGA TCT GAT ACG GAT CTG GCC-3′. As a control, we used GAPDH gene with forward primer: 5′-CAA GGT CAT CCATGA CAA CTT TG-3′, and reverse primer: 5′-GTC CAC CAC CCT GTT GCT GTA G-3′. We normalized the number PCR cycle threshold C_t_ for the target sequence to the one for GAPDH control gene. Experiments are performed in triplicate.

### 2.5. In vivo Experiments

#### 2.5.1. Animal Experimentation

Animal experiments were performed in accordance with the ethical project submitted and approved by the ethical committee N°26 regulating the animal facility at Gustave Roussy Institute (Villejuif, France) and under national agreement N°2013-062-01223.

#### 2.5.2. Biodistribution of Nanodiamonds in Mice

We injected 100 µL of PBS buffer containing 3 × 10^6^ A673 cells in the right flank of male nude mice. When the tumors appeared, we injected in the tail veins of the mice 100 µL of water for injectable preparation containing either siRNA (5 mg/kg) alone, T-DND (25 mg/kg) alone or siRNA complexed to T-DND (5 mg/kg siRNA mixed to 25 mg/kg T-DND, i.e., mass ratio: 5:1). We then placed the mice in metabolic cages for 4 h or 24 h. At these times, 3 mice of each condition were humanely sacrificed, the tissues were withdrawn and their weights are determined. About 0.1 g of each tissue is sampled and solubilized with 1 mL of Solvable™ (PerkinElmer, Courtaboeuf, France) heated during 3 h at 55 °C. After the transfer of the solubilized tissues in scintillation tube, solutions were clarified by adding 100 µL of 30% hydrogen peroxide (Sigma-Aldrich S.a.r.l, Saint-Quentin Fallavier, France). Then 10 mL of Ultima Gold (PerkinElmer, Courtaboeuf, France) were added before the radioactivity measurements were performed during 1 min with 1409 DSA scintillation counter (Wallac/PerkinElmer, Courtaboeuf, France). The results are expressed as deionized counts per minutes per unit mass of tissue (cpm/g) or as the percentage of the injected dose to the whole organs. All urines from cabinet containing animal group receiving a same treatment (in these conditions, measurement is a global statistical value) were collected. Experiments were carried out in triplicates for each condition. For a healthy mouse, the glomerular filtration rate of a compound intravenously injected is about 7 µL/min per g, leading to an almost full elimination after 3 h [23]. Therefore, at our observation times of 4 h and 24 h, we should be able to detect T-DND in urines, if they could be eliminated through this pathway.

#### 2.5.3. Measurement of *EWS-FLI1* Inhibition in Nude Mice

*EWS-FLI1* expression in tumor was measured after intravenous injection in the tail vein of nude mice of 100 µL of antisense or control siRNA 5 mg/kg bound to H-DND at mass ratio of 5:1 (H-DND:siRNA) or free H-DND for 24 h. Then a fragment of the tumor was sampled and placed in a tube containing 400 µL of Trizol solution (Invitrogen, Life Technologies S.A.S., Courtaboeuf, France). A ceramic ball was added and tissue homogenization was performed for 30 s in TissuLyser II (Quiagen Paris, Courtaboeuf, France). Finally, *EWS-FLI1* expression was quantified by RT-qPCR as previously described.

### 2.6. Statistical Analysis

Data are presented as means and either standard deviation or standard error on the mean. Statistical significance was evaluated using GraphPad Instat 3.10 software (GraphPad Sofware, San Diego, CA, USA). For all the comparisons between two conditions, we used the non-parametric Wilcoxon–Mann–Whitney two sample rank test (two-tailed *p* value provided), while for the comparison between more than three conditions we applied a Kruskal–Wallis test, followed by Dunn’s Multiple Comparisons Test. One star (*) corresponds to *p* < 0.05, ** to *p* < 0.01 and *** to *p* < 10^−3^.

## 3. Results

### 3.1. Characterization of the Hydrogenated and Tritiated Nanodiamond Suspensions

After hydrogenation or tritiation by annealing using molecular gases (see methods and [21]), we suspended the treated DND powder in distilled water by sonication. We measured hydrodynamical diameters of DND clusters ranging from about 60 nm to about 90 nm for H-DND and T-DND colloidal suspensions, respectively (Appendix A). These results are in agreement with previous colloidal studies on hydrogenated DND [21,24] and also with the transmission electron microscopy observations we reported in Ref. [16] (see Figure 4a in this reference), showing H-DND aggregates of ≈50 nm in size at the cell membrane.

### 3.2. Loading Capacity of Nanodiamonds for siRNA Binding

We quantified the loading capacity of H-DND for siRNA binding. To this aim, we measured by fluorescence (see Materials and Methods), for a fixed amount of siRNA, the remaining free unbound siRNA for increasing amounts of hydrogenated nanodiamonds. The results are presented in Figure 1 for two conditions: one in which H-DND and siRNA were simply mixed, the other in which the mixing was done under sonication, both for the same duration. For these two conditions, the complete fixation of siRNA occurred at the same mass ratio *m*_H-DND_/*m*_siRNA_ of 10:1. These data suggest the sonication assistance leads to a faster adsorption of siRNA onto H-DND, resulting from the larger thermal agitation brought by ultrasound waves.

### 3.3. Effect of siRNA Binding to Nanodiamonds on the Hydrodynamic Size and Zeta Potential of Complexes

Figure 2 displays the variation of the hydrodynamic size and zeta potential of H-DND:siRNA complexes upon increase of the DND:siRNA mass ratio. The experiment was carried out under two conditions: with or without sonication during complexation. We observed that for mass ratios lower than 10 the particles hydrodynamic size is around 80 nm (Figure 2a). Then, the size increased strongly for mass ratio between 10 and 40 to return to the initial size for mass ratio higher than 40. We conclude that strong aggregation occurred only for mass ratios between 10 and 40.

Looking at the zeta potentials of each solution (Figure 2b), we observed that for mass ratios lower than 10, the surface charge is negative (around −30 mV) corresponding to an excess of siRNA covering nanodiamonds. For mass ratios between 10 and 40, a surface charge inversion occurs with a zeta potential close to +40 mV for a mass ratio of 40. In between, the low zeta potential of the complexes promotes their aggregation, as observed on Figure 2a. For mass ratios higher than 40, the positive zeta potential reflects an excess of hydrogenated nanodiamonds. Sonication have no effect on surface charge but promote smaller aggregates, as observed on the DLS profiles (Figure 2a).

Moreover, considering that we inject the H-DND:siRNA conjugate in the blood, we could have some concern regarding its aggregation in such a complex environment. However, it is well established (see for example S. Hamelaar et al. [25]) that the serum favors the dispersion of electrostatically-charged nanoparticles when they are dispersed in a culture medium. In order to confirm these data with our specific conjugate, we formed a solution of H-DND:siRNA at the same mass ratio of 5:1 used in the in vivo experiment, and measured its hydrodynamic diameter in DMEM culture medium supplemented with 10% Fetal Bovine Serum. We found a diameter of 35 nm, a value even smaller than the diameter of 90 nm of the same conjugate in pure water (see Appendix A), which is a strong indication that injecting intravenously an aqueous suspension of H-DND:siRNA (mass ratio 5:1), will rather induce a small deagglomeration than an aggregation in the blood circulation.

### 3.4. Inhibition of EWS-FLI1 on Ewing Sarcoma Cultured Cells

We first studied the efficacy of H-DND prepared by annealing under hydrogen gas, as vector for siRNA delivery to human Ewing sarcoma A673 cells. We used different mass ratios between H-DND and siRNA, from 5:1 to 50:1. We observed on Figure 3 that 35% inhibition is obtained with these cationic H-DND for mass ratios higher than 25. In comparison, commercially available cationic liposomes Lipofectamine 2000 used in similar conditions in the presence of serum during transfection show a lower efficacy with only 18% inhibition. It is to note that in recommended conditions (transfection in serum free medium) Lipofectamine 2000 yields 70% inhibition on this cell model [9]. The efficacy of siRNA transfection by H-DND in serum containing medium is crucial for further applications in animals.

### 3.5. Inhibition of EWS-FLI1 Expression in Mice

We then evaluated the efficacy of siRNA vectorized by T-DND to inhibit *EWS-FLI1* in tumor xenografted on mice. We produced *EWS-FLI1* tumor model by grafting A673 cells in the flank of nude mice. We used tritiated DND to conduct both the inhibition and biodistribution studies using the same mice. We selected the mass ratio of 5:1, first of all to ensure that all T-DND are covered by siRNA. Our hypothesis was that such a configuration would favor the detachment of siRNA molecules which interact only by a portion of their total length with the T-DND surface. Moreover, the mass ratio of 5:1 also ensures that the injected suspension is not too viscous, since at the given necessary dose of siRNA and volume of solution to be injected, the concentration of T-DND remains low enough to have a small impact on the viscosity of the aqueous suspension. Finally, high dose of cationic vector may be toxic as observed in cultured cells. For all these reasons, we favored the smallest H-DND concentration, corresponding to the 5:1 mass ratio.

We injected the T-DND radiolabels in the tail vein of mice, using either uncomplexed T-DND or with siRNA complexed to them. We did not include a free siRNA group because we already established that free siRNA is not able to inhibit *EWS-FLI1* oncogene [26]. We considered 6 to 8 nude mice per condition, which were placed in metabolic cage. After 4 h or 24 h, mice were humanly sacrificed. For the efficacy study, we only considered the 24 h group. Using RNA extracted from tumors, we then quantified *EWS-FLI1* expression by RT-qPCR. The normal level of *EWS-FLI1* expression is provided by mice treated by non-complexed T-DND. We also used an irrelevant siRNA complexed with T-DND as a specificity control. We observed on Figure 4 that the irrelevant T-DND:siRNA treatment has no effect. The *EWS-FLI1* antisense siRNA complexed to T-DND inhibits the *EWS-FLI1* expression by about 50%. These results confirm that H/T-DND:siRNA is able to inhibit *EWS-FLI1* in this Ewing sarcoma tumor model grafted on nude mice.

### 3.6. Biodistribution of Nanodiamonds in Nude Mice

For an efficient use of H-DND produced by annealing for siRNA delivery to Ewing sarcoma tumor xenografted on mice, it is important to determine their tissue distribution and elimination, which is made possible by radioactivity measurement using their radioactive analogs T-DND. The different tissues of the mice of the efficacy study (Section 3.5) were collected, homogenized and then the radioactivity was determined as presented in Figure 5. We also collected urines and feces at the same time points. We observed that T-DND accumulated mainly in the liver, lung, spleen, kidney and also the heart for T-DND:siRNA. There are no significant changes between 4 h and 24 h in the quantity of T-DND found in the different tissues. For kidney, spleen, lung and heart, T-DND complexed with siRNA accumulated more than uncomplexed T-DND. The radioactivity is recovered in all tested tissues at a lower level. We did not observe high accumulation in the tumor. Nevertheless, the dose measured in the tumor is four time larger than the one in the blood.

One of the major questions for the use of mineral origin nanoparticles for biomedical applications is how they may be eliminated after injection in animals. The radioactive fraction recovered in urines was 0.15% of the injected dose per mouse. Unfortunately, after centrifugation of urines, we found that all radioactivity was recovered in the supernatant. This indicates that no T-DND was eliminated through the kidney pathway. The very small detected radioactivity is probably the one of water consecutive to exchange of H with labile T. Note that 4 h after injection in the kidney we detected about 0.25% of the radioactive T-DND:siRNA injected dose. This can be due either to the smallest T-DND being filtrated by the glomeruli but then reabsorbed by the tubules and/or to T-DND aggregates not being able to cross the glomeruli. This last hypothesis is in agreement with the fact that uncomplexed T-DND accumulated about twice less in kidney than the one associated to siRNA, which present a higher aggregation state. Another possible elimination pathway is by the feces. Thanks to the metabolism cage, we estimated that the total collected amount of feces after 24 h had a radioactivity representing about 0.19% for H-DND:siRNA and 0.13% for H-DND of the injected doses per mouse (Figure 5b), which is about 6 times larger than the one after 4 h (0.03% and 0.02% for H-DND:siRNA and H-DND, respectively), indicating a linear elimination with time, in perfect agreement with the constant value of radioactivity in feces sampled in the rectum at 4 h and 24 h (Figure 5a). Indeed, since the majority of DND are captured by liver cells, their transfer to the intestine lumen by the bile may be the main way for T-DND elimination.

## 4. Discussion

In a previous work, we had demonstrated that cationic hydrogenated detonation nanodiamonds can deliver active siRNA to Ewing Sarcoma cell in culture [16]. Here, we wanted to establish if small hydrogenated DND are able to deliver active siRNA in vivo to mice model of Ewing sarcoma. In order to investigate simultaneously their biodistribution, we produced hydrogenated and tritiated detonation nanodiamonds via their exposure to a mixture of H_2_ and T_2_ molecular gases at 550 °C under controlled pressure. The characterization in size and surface charge was performed by DLS measurement and zeta potential determination (Appendix A). We observed that H-DND are cationic with a zeta potential of ≈+45 mV and have a hydrodynamic size of ≈60 nm. H-DND positive charge favored electrostatic attachment of siRNA onto the nanoparticles and the measured DLS size larger than the 7-nm primary size indicated some aggregation. This size difference is partly due to the measurement methods, as unitary diamond size is the one of its hard core as observed by high-resolution transmission electron microscopy, whereas DLS yields the hydrodynamic size, that is larger because it takes into account the water dipole molecule attached to the core and moving with it under Brownian motion. Our data indicate that H-DND are present in the aqueous suspension in the form of aggregates of an average number of eight primary 7 nm sized particles.

The binding capacity of H-DND was studied by mixing to a constant quantity of siRNA an increasing quantity of H-DND and the full binding of siRNA was obtained for 10 times excess of DND (Figure 1). Hydrodynamic size and zeta potential variations during the titration process of siRNA with H-DND (Figure 2a) indicated that for mass ratio around this point of saturation (in the 10 to ≈40 mass ratio range) the size of H-DND:siRNA complexes increased to form large aggregates. The sonication during siRNA and H-DND association process decreased the size of the aggregates but did not prevent it. For mass ratio smaller than 10 or larger than ≈40 the nanoparticle hydrodynamic sizes were close to the ones of uncomplexed H-DND (<100 nm, Appendix A). Furthermore, the slightly smaller size of the H-DND:siRNA complex under sonication indicates its high colloidal stability. This strong binding capacity of H-DND presents advantages for animal applications to prevent desorption of the cargo before complexes reach their target cells. Finally, during the titration process, the zeta potential decreased from +50 mV for an excess of H-DND (mass ratios larger than 20) to a negatively charged complexes (zeta potential ≈-30 mV) for mass ratios smaller than 10 (Figure 2b). The sonication did not change the mass ratio transition threshold. This observation is consistent with the aggregation happening once all nanodiamonds have lost their charges due to the complexation with siRNA. It is worth to note that the global charge of the complexes could be tuned to negative (mass ratios smaller than 10) or to positive (mass ratios larger than 20), resulting in the formation of complexes that may interact differently with cell membrane and lead to different siRNA delivery efficacy [27].

In the prospect of using H-DND for anticancer gene delivery in mice, we first validated the delivery of these cationic DND by their capacity to transport a siRNA cargo directed against *EWS-FLI1* oncogene in cultured A673 human Ewing sarcoma cells and by detecting a gene inhibition efficacy. We observed that H-DND vectorized siRNA inhibited *EWS-FLI1* expression by 30–40% (Figure 3). In comparison with previous results [16], the inhibition of *EWS-FLI1* expression by siRNA transported by DND hydrogenated via annealing is lower than the one obtained with plasma hydrogenated DND vectors. This may be due to the slightly larger aggregation of DND hydrogenated via annealing, leading to a lower internalization efficiency [28].

One key question regarding the use of non-biodegradable nanoparticles for drug delivery in animal is to determine whether they are eliminated, through which pathways, and where in the body the non-eliminated fraction accumulates. It was shown that “large” ND of 50 nm primary size reside in animal tissues a few weeks after injection [17]. However, “small” ND of size compatible with kidney elimination may overcome this limitation. Indeed, nanoparticles smaller than 6–8 nm have been already reported to be filtered at the glomeruli level [18,19]. In this respect, detonation ND with primary size included between 3 and 8 nm and even smaller [29] are good candidates for such elimination. For this study, we prepared radioactive tritiated DND to be able to trace these ultra-small particles after injection to mice. We observed that the liver, spleen and lung contained much more radioactive amount than the other tissues (Figure 5). Moreover, T-DND carrying siRNA accumulated more in kidney, spleen, heart and lung, with a slow increase from 4 to 24 h. Unfortunately, we observed a limited accumulation of T-DND in the tumor compared to other tissues such as muscles, heart or kidney. This indicated that Enhanced Permeability and Retention (EPR) [30] effect did not occur for these complexes or/and that the tumor had no aberrant vascularization responsible for EPR effect. Nevertheless, T-DND:siRNA quantity was stable in the tumor between 4 h to 24 h which is the time range needed to induce the silencing effect. During the first 24 h, the blood radioactivity did almost not decrease, which indicated that DND were only very slowly eliminated from the circulation.

The main possible elimination pathways are urines and feces. Radioactivity from urines could come from both T-DND or tritiated water. The later could originate from a fraction of tritium desorbing from T-DND and exchanging with hydrogen in water. Using ultracentrifugation, we measured this labile tritium fraction to be 5% (see Appendix A). In order to reduce the amount of free tritium in the injected solution, all T-DND:siRNA complexes were prepared from T-DND purified by this ultracentrifugation process. However, even after such purification, there is a remaining fraction of 2% free tritium. This value provides a lower bound of T-DND detection sensitivity in urines. Urines were collected globally for each group of mice and their radioactivity was measured in 1 mL of solution and in 1 mL of ultra-centrifugated supernatant. We observed the same values of radioactivity before and after centrifugation (Appendix A) within a margin of error larger than the detection sensitivity lower bound. Hence, the radioactivity found in urines most likely comes from the residual free tritium present in the injected solution and not from T-DND. We therefore consider that T-DND were not eliminated in urines, or only very slowly. The possible reasons for the absence of T-DND in urine are (i) the formation of aggregates too large to be filtrated or/and (ii) ultrasmall DND reabsorbed by kidney tubules, the first one (i) being the most likely, considering previous observations of Rojas et al. [20] who showed that membrane-filtrated DND are eliminated in urines, much more efficiently than the aggregates dominant in the original suspension.

We then explored the other possible elimination pathway, by collecting the feces from the rectum and we measured their radioactivity. We observed a high radioactive signal in the feces. The amount of radioactivity in the feces fraction is larger than the one of free tritium in the injected volume, according to the labile tritium release measurements (Appendix A), therefore we shall conclude that DND are present in the feces which therefore constitute one of the elimination pathways. It is likely that DND are eliminated though the bile by liver filtration. Since the bile goes into the intestinal lumens, DND are then incorporated to the feces. This result is of high importance because it states that ultrasmall DND can be partly eliminated from mouse body after intravenous administration.

Furthermore, if we consider the radioactivity balance between injected doses per mouse and the total amount collected, only about 25% of the dose is recovered in the sampled fractions. The 75% of not-measured radioactive T-DND are localized in the carcass containing skin, not removed muscles, bones and so on, and probably a part has also been eliminated through other routes. Indeed, the elimination pathways are multiple in animals. Some authors propose expectoration as an alternative way of elimination. Considering that DND also accumulated into the lung (Figure 5), they may be engulfed by macrophages which further direct them into the alveoli where they may be finally expulsed through the pharynx by the mucociliary system [31]. Further investigations would be necessary to evaluate the expectorated fraction.

Finally, regarding the silencing efficacy of *EWS-FLI1* in Ewing sarcoma cells xenografted tumor by siRNA delivered by T-DND, we observed a 50% inhibition by the antisense siRNA compared to irrelevant siRNA. The latter yielded a similar effect than the T-DND alone, i.e., no inhibition. This result indicates that despite a low accumulation of T-DND in the tumor, the siRNA is delivered to the cancer cells where they silence the targeted oncogene. Note that although the in vitro inhibition efficacy is similar to the in vivo one, one cannot generally extrapolate in vitro results to in vivo ones which justifies our study. Finally, in order to treat the mouse and reduce the tumor size, we could combine a conventional anticancer treatment like vincristine to the H-DND-siRNA one, as we showed in our previous in vitro study [16].

Overall, our study represents a significant step towards the use of ultra-small solid nanoparticles which are able to deliver efficiently active siRNA to tumor cell in animals and are also eliminated from their body subsequently.

## Figures and Tables

**Figure 1 nanomaterials-10-00553-f001:**
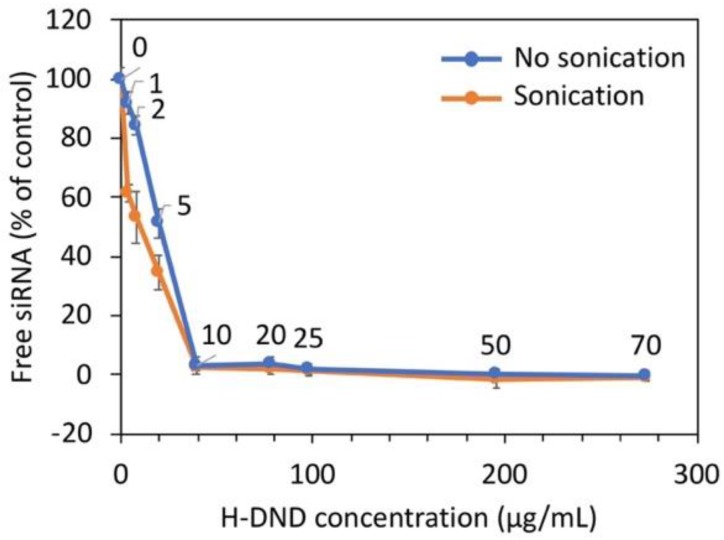
Binding of siRNA to hydrogenated detonation nanodiamonds (H-DND). Free siRNA as function of increasing quantity of H-DND (H-DND-22 samples) added to a fixed initial quantity of siRNA. The results are presented as the percentage of initial siRNA. Orange/blue with or without (respectively) sonication during the siRNA to H-DND complexation. Experiments were performed in triplicates. Number over each point represent the mass ratio *m*_H-DND_/*m*_siRNA_. The statistical comparison tests between sonicated and non-sonicated conditions yield *p* = 0.021 (*) for mass ratio 1:1 to 5:1 and non-significant differences starting from mass ratio 10:1 to higher ones.

**Figure 2 nanomaterials-10-00553-f002:**
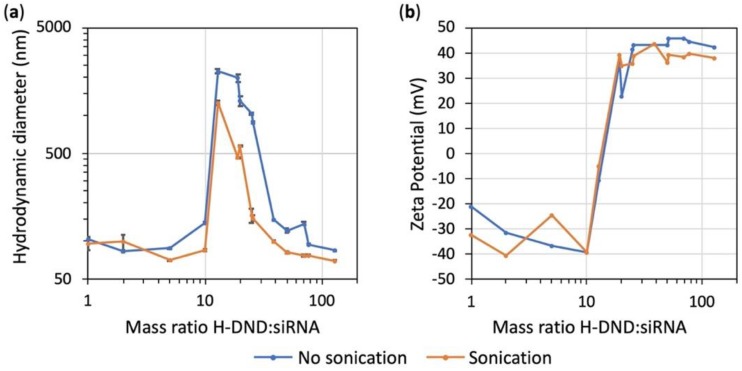
Variation in hydrodynamic size (in intensity) and zeta potential of siRNA bound to increasing mass ratio of H-DND (H-DND-22 sample). Complexes were prepared with or without sonication. (**a**) Size measurement by dynamic light scattering (DLS). (**b**) Zeta potential determination.

**Figure 3 nanomaterials-10-00553-f003:**
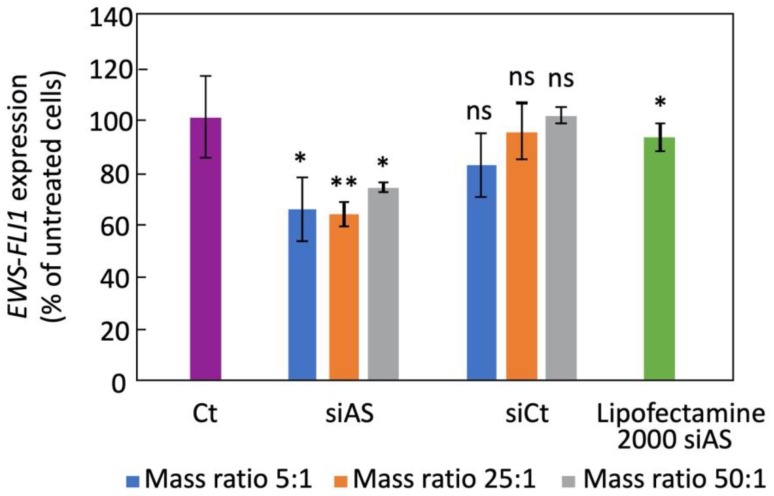
Inhibition of *EWS-FLI1* expression in A673 Ewing sarcoma cultured cells by siRNA vectorized by H-DND (sample H-DND-24) or Lipofactamine 2000. Cells are treated during 24 h in DMEM medium containing 10% of fetal calf serum by 75 nM siRNA. The H-DND:siRNA mass ratio was 5:1, 25:1 or 50:1. siAS: antisense siRNA directed against *EWS-FLI1* oncogene; siCt: control irrelevant siRNA sequence (see methods). Experiments were performed in triplicate. Comparisons were all done relative to the “Ct” condition using the Wilcoxon–Mann–Whitney two sample rank test, that yielded *p*_5:1_ = 0.018, *p*_25:1_ = 0.009; *p*_50:1_ = 0.036, *p*_lipo_ = 0.019; *ns*: non-significant.

**Figure 4 nanomaterials-10-00553-f004:**
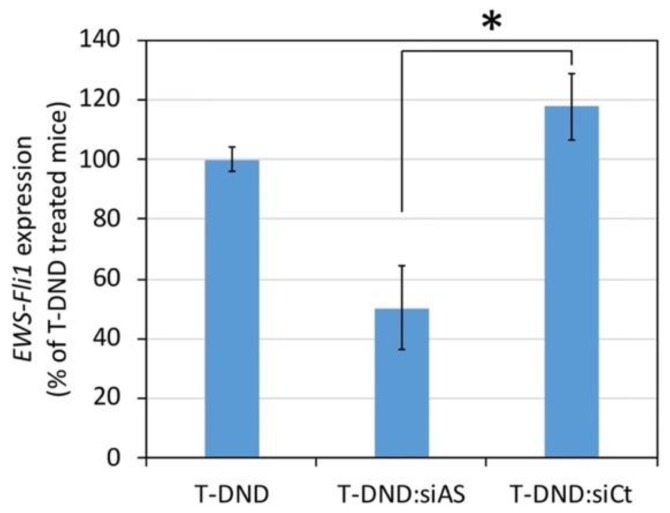
Inhibition of *EWS-FLI1* expression in tumor xenografted on mice treated for 24 h by siRNA vectorized by tritiated DND (T-DND) at a mass ratio of 5:1 (T-DND:siRNA). The tumor was formed from A673 Ewing sarcoma cells grafted in the right flank of the mice (*n* = 6 to 8 animals per condition). T-DND:siRNA was intravenously injected in the mouse tail vein. RNA was extracted from the tumor of mice sacrificed 24 h after treatment, and RT-qPCR was performed using GAPDH as housekeeping gene. Standard error on the mean are indicated. siAS:siRNA against *EWS-FLI1* oncogene; siCt: irrelevant. A significant difference between T-DND:siAS and T-DND-siCt is observed (*) according to the *p*_Dunn_ value <0.05, inferred from a Kruskal–Wallis test (*p* = 0.0094) followed by a Dunn multiple comparisons test.

**Figure 5 nanomaterials-10-00553-f005:**
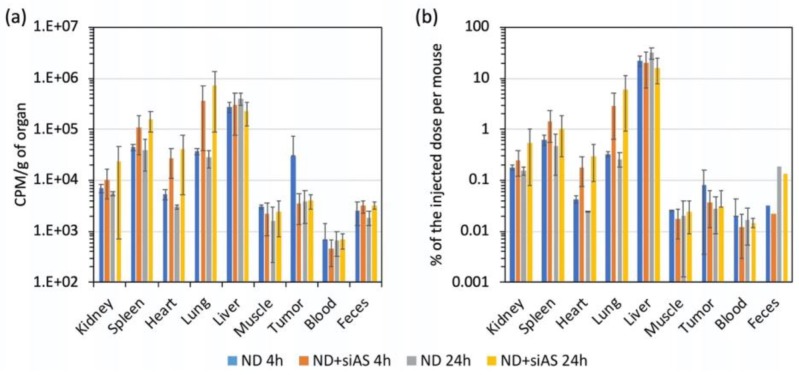
Mouse organ distribution of uncomplexed T-DND or complexed with siRNA at mass ratio 5:1 of T-DND:siRNA. The total radioactivity of each tissue is measured after 4 h and 24 h for 3 animals per series. Error bars: standard deviations. (**a**) Radioactivity presented as count per minutes (cpm) per gram of tissue or feces. The latter corresponds only to the fraction collected in the rectum of each animal. (**b**) Radioactivity from each full organ, presented as the percentage of the total injected dose. Feces were collected for all the animals simultaneously in the same metabolism cage, but the total amount, including non-expelled feces is not known which is why we could not include feces data in B. ND + siAS: DND complexed with siRNA sequence antisense for *EWS-FLI1* oncogene. The differences between error bars for the different conditions are due to (i) the variability between animal groups, and (ii) to the fact that the T-DND quantifications were done only on portions of ≈100 mg of each organ, and that we cannot exclude an inhomogeneous distribution of T-DND within the organs.

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
