# Peer review of "Delivery of siRNA to Ewing Sarcoma Tumor Xenografted on Mice, Using Hydrogenated Detonation Nanodiamonds: Treatment Efficacy and Tissue Distribution"

_nanomaterials, 2020, doi:10.3390/nano10030553_

Round 1

Reviewer 1 Report

The study by Claveau et al. investigates the use of hydrogenated detonation nanodiamonds as a cargo for EWS-FLI1 specific siRNA for the treatment of Ewing sarcoma in a xenograft mouse model. The authors used tritiated nanodiamonds to trace their in vivo districtibution.

The authors show that these nanodiamonds enrich in lung and liver and only to a minor fraction in tumor tissue. Here they show a better downregulation of EWS-FLI1 mRNA by the diamonds loaded with the specific siRNA than the controls.

This is an interesting study but the reviewer has difficulties to understand what do we learn from this study.

However, it is unclear what affects clearance of these diamonds 

  1. Is the size of the hydrogenated detonation nanodiamonds to big for a good clearance via the kidney?
  2. is the investigated time period to short for a good/significant clearance of these nanodiamonds?

Has the enrichment in lung, liver and other tissues toxic side effects?

Is there a therapeutic value (tumor shrinkage compared to low tissue toxicity)?

Minor: The translocation product should be written EWS-FLI1.

Author Response

We thank the reviewer for his interest in our study and for his suggestions of improving the clarity of the manuscript.

Here are our replies (R) to each question (Q):

Q: “Is the investigated time period too short for a good/significant clearance of these nanodiamonds”

R: No, the urines have been collected during 24 hours after the DND-based nanoconjugate injection, while in a healthy mouse the glomerular filtration rate (of about 7 µL/min per g [see Y. Sasaki et al, Phys. Rep. 2, e12135 (2014), doi: 10.14814/phy2.12135]) should ensures elimination of intravenously injected products within about 3 hours [10.14814/phy2.12135]. So, our observation time was long enough to detect urinal clearance. We added this precision and the reference to the revised manuscript, in the section 2.4.2. (Biodistribution) of the Materials and Methods.

Q: “Is the size of the hydrogenated detonation nanodiamonds too big for a good clearance via the kidney?”

R: In principle, with a DND primary size of 7 nm, the hydrogenated detonation nanodiamonds (H-DND) we used can be excreted in the urines. This has been demonstrated by Rojas et al., ACS Nano 5, 5552 (2011) using 18F-radiolabeled DND that were membrane-filtered (reference added to our revised manuscript, Ref. 20). Our data (Fig. 2(a)) show that uncomplexed H-DND of H-DND complexed with siRNA (at mass ratio<10 or >40) reaggregate, even in pure water, leading to hydrodynamical size of 60-90 nm (from DLS measurement, in intensity). Although these aggregates can be disassembled by sonication, we believe that the absence of urinal clearance indicates that the kidney cannot provide sufficient energy for DND deagglomeration. We have insisted more on this conclusion in the discussion (lines 591-593) of the revised version. Moreover, like in Rojas et al., we could have filtered the H-DND:siRNA complex though a membrane. However, to inject the dose of siRNA which is high enough to inhibit the oncogene expression we would have had to re-concentrated the filtrates by centrifugation. Such a procedure carries the risk of a re-agglomeration.

Q: “Is there a therapeutic value (tumor shrinkage compared to low tissue toxicity)?”

R: The oncoprotein EWS-FLI1 favors tumor growth, but stopping its production is not sufficient to reduce the tumor size by itself. Indeed, we did not observe a tumor shrinkage 24 h after injection of the H-DND:siRNA. To achieve tumor size reduction, the siRNA strategy needs to be combined with an anticancer treatment like vincristine, as we have shown in our previous in vitro [J.-R. Bertrand el al., Biomaterials 45, 93 (2015), Ref. 16]. We have added this strategy in the conclusion (lines 617-619). Hence our study does not provide the demonstration of a therapeutic value but it constitutes a first step, demonstrating the ability of delivering siRNA being active in vivo.

Q: “Has the enrichment in lung, liver and other tissues toxic side effects?”

R: We did not observe any animal behavior changes during the 24 h before the sacrifice, for any of the compound (either H/T-DND alone or complexed with siRNA). We neither detected organ abnormalities by visual inspection during their sampling. However, a longer observation time, with more quantitative evaluation (e.g. dosage of creatinine in urines to detect renal dysfunctions; dosage of alanine transaminase, aspartate transaminase, and gamma GT for liver failures), would have been necessary to achieve a solid conclusion regarding the animal toxicity.

Q: “What do we learn from the study”?

R: To our knowledge there is a single other study than ours reporting the tissue distribution of DND in mice after intravenous injection, i.e. the one of Rojas et al. ACS Nano 2011 Ref. 20. In the later, the authors used hydroxylated DND further functionalize with amino groups, and found in the case of mice very different tissue distribution results than ours: their DND aggregates (340 nm in size) accumulated ≈20 times more in the lungs than in the liver, which is an unexpected result if their injection was done in the vein of the tail (which is not mentioned) but could be compatible with a retroorbital injection. Hence, we believe that our study complements the only previous and not fully documented DND biodistribution investigation. Furthermore, as stated earlier, it reports the first successful in vivo delivery with DND (injected intravenously) of an siRNA active in inhibiting a targeted gene, as stressed in the conclusion.

Q: “The translocation product should be written EWS-FLI1”

R: We replaced the spelling EWS-Fli1 by EWS-FLI1.

Reviewer 2 Report

In this report by Claveau et al. the authors present an extension on their previous work on the preparation of cationic detonation nanodiamonds (DND) (Ref 16) and their tritium labelling (Ref 20). While the approach is of interest, I have several concerns about the present work. Given the material, siRNA loading and tritium labelling are previously published, the novelty of this report lies in the preclinical assessment of such nanomaterials, however I have reservations about this portion of the report;

  • The stability of the radiolabel in biological conditions (in blood or at least serum) is of key concern to the ability of the study to produce meaningful outcomes. Given the lack of such stability studies it makes it difficult to draw conclusions from the presented biodistribution study and the presented interpretations may be spurious (for example the lack of labelled particles detected in urine, and the postulation of no EPR effect).
  • In the presented biodistribution study, a DND:siRNA ratio was chosen which appears to form large aggregates (as suggested by high lung accumulation for example) and would not achieve complete binding of siRNA (Fig 1). Why was this ratio chosen and if the actual assembly of siRNA complexes are not discrete and easily characterized, can meaningful conclusions be drawn from this study? Or compared to the non-siRNA control particles?
  • In the therapeutic portion of this work a control arm comprised of free siRNA therapy would be of interest to compare to the DND delivered materials

In addition to these reservations I also had some minor queries about other portions of the work;

  • In the DND:siRNA complexation experiments sonication is noted to make a small difference (Fig 1). Is this difference statistically significant at any of the ratios assessed? And is this difference due to altered kinetic of association and could this be assessed in a dynamic assay?
  • In Fig 3 are any of the differences in expression statistically significant? If so these should be emphasised as this would enhance the argument for effective delivery
  • In Fig 4 why are the error bars on the measurements non-uniform?

In conclusion while the study presents interesting results, I believe publishing in it’s current form would be premature at this stage. The manuscript could be significantly strengthened if these issues are able to be addressed.

Author Response

We thank the reviewer for his interest in our study, and for the questions he raised, giving us the opportunity to clarify some aspects of our work in the revised version.

Replies to specific concerns:

Q: “The stability of the radiolabel in biological conditions…” and “spurious interpretation of the results”; “…can meaningful conclusions be drawn from this study?”

R: Assuming that by “radiolabel” the reviewer means T/H-DND, the stability concern he is referring to is probably the one of its size in the serum containing medium or in blood. However, it is well established (see for example S. Hamelaar et al., Microchimica acta 184, 1001 (2017), added Ref. 25) that the serum favors the dispersion of electrostatically-charged nanoparticles when they are added to a culture medium. The strong ionic force of the later would otherwise screen the charge repulsion that ensure the nanoparticle colloidal stability in pure water, but the protein contained in the serum (e.g.bovine serum albumin) rapidly form a corona around the DND, maintaining them apart from each other. The conjugate (T/H-DND:siRNA) was prepared in pure water, and this is also the same aqueous suspension that was injected in the mice vein. During injection, when the conjugates are put in contact with the blood serum, the proteins it contains adsorb on their surface and prevent them to aggregate, leading to a size that can be even smaller to the one in pure water. In order to confirm such a scenario, we measured the size of the H-DND:siRNA complex in a serum containing culture medium. We found a diameter of 35 nm, a value even smaller than the diameter of 90 nm of same conjugate in pure water (see Table S1), which is a strong indication that injecting intravenously an aqueous suspension of H-DND:siRNA (at mass ratio 5:1), will rather induce a small deagglomeration than an aggregation in the blood circulation. We added these additional measurements (H-DND:siRNA diameter in water and serum supplemented culture medium) to Table S1.

Hence, considering the high stability of the complex in the serum containing medium, our study was indeed realized in well-controlled size conditions. We have added these precisions and the reference S. Hamelaar et al. 2017 (Ref. 25), at the end of the paragraph 3.3 (Effect of siRNA binding to nanodiamonds on the hydrodynamic size).

Q: Why did we choose the H-DND:siRNA mass ratio of 5, while it appears to form large aggregates (as suggested by high lung accumulation for example) and would not achieve complete binding of siRNA (Fig 1)?

R: First of all, at the mass ratio of 5:1, the hydrodynamic size of the H-DND:siRNA complex is smaller than 100 nm according to Fig.2(a). The reviewer extrapolation: “the complex appears to form large aggregates (as suggested by high lung accumulation for example)”, based on the biodistribution data is not supported by our in vitro measurements (Supp. Info. Table S1). The tissue accumulation results from multiple processes that does not depend only on the size of the conjugate at the place of injection.

Then, the choice of a mass ratio of 5:1 for which siRNA is in excess during the titration process and is therefore present as unconjugated free siRNA in the suspension (Fig.1, as pointed out by the reviewer), is to ensure that all H-DND are covered by siRNA. We believe that such a configuration favors the detachment of siRNA molecules which interact only by a portion of their total length with the H-DND surface.

Moreover, the mass ratio of 5:1 also ensures that the suspension injected is not too viscous, since at the given necessary dose of siRNA and volume of solution to be injected, the concentration of H-DND remains low enough to have a small impact on the viscosity of the aqueous suspension.

Moreover, in our previous work (Ref. 16) we showed that H-DND cationic nanoparticles are toxic at “high” concentration, which is another reason to favor the loss mass ratio of 5:1, in order to limit a possible toxicity for in vivo experiments.

We thank the reviewer for having pointed out this lack of justifications. We added them to the section “Inhibition of EWS-FLI1 expression in mice”. So, at mass ratio of 5:1, the conjugate is not only well characterized (for both the antisens siRNA or the control one), but it is also well adapted to an efficient in vivo delivery of siRNA as confirmed later by the inhibition results.

In order to strengthen more this rational choice of mass ratio for an efficient delivery, we switched the presentation of the inhibition efficacy and the biodistribution: now section 3.5 is “Inhibition of EWS-FLI1 expression in mice”, and Fig.5 becomes Fig.4 and vice-versa. This required some minor adjustments of these sections which did not affect the substance of the text.

Q: “In the therapeutic portion of this work a control arm comprised of free siRNA therapy would be of interest to compare to the DND delivered materials”

R: It has already been established that free siRNA is not able to inhibit EWS-FLI1 oncogene (see in particular Toub et al., Pharmaceutical Research 23, 892 (2006), added Ref. 26). This is due to its rapid degradation by nuclease, especially in vivo. This is the reason why we did not use free siRNA in our study. We added this comment and the reference (Ref. 26) in section 3.5.

Furthermore, siRNA can also trigger immune response (in particular the complement) when delivered unconjugated to a nanoparticle, and it will therefore not have constituted a valid control.

Minor queries:

Q: Fig. 1: “In the DND:siRNA complexation experiments sonication is noted to make a small difference (Fig 1). Is this difference statistically significant at any of the ratios assessed?”

R: For each mass ratio between 1:1 and 10:1 we did three measurements, and a Wilcoxon rank test comparison between “with” or “without sonication” yielded p=0.021 (* significant) for the ratios 1:1 to 5:1 included, and p=0.420 (non-significant) for the ratio 10:1. These results confirm that the sonication facilitates the adsorption of siRNA onto H-DND. We added the statistical significance test results to Fig.1 caption.

Q: Fig. 1: “Is this difference due to altered kinetic of association and could this be assessed in a dynamic assay?”

R: The facilitated association of siRNA to H-DND in the presence of sonication, leads to the same final state of a complete fixation at mass ratio of 10:1 than without sonication. These observations are in favor of a faster kinetics of association under sonication resulting from the larger thermal agitation brought by ultrasound waves. This explanation could be checked by varying the time of interaction of siRNA with H-DND in order to infer the kinetic constants, but such experiments would be out of the scope of our study. To take into account the reviewer’s comment, we slightly modified the conclusion of section 3.2.

Q: “In Fig 3 are any of the differences in expression statistically significant? If so these should be emphasised as this would enhance the argument for effective delivery”.

R: We thank the referee for pointing out this lack of precision: yes, there are significant differences between “Ct” and “antisense siRNA/siAS” and not between “Ct” and all the other conditions. We have added the p values to the revised version of Fig. 3 and its caption.

Q: “In Fig 4 why are the error bars on the measurements non-uniform?”

R: The reasons of the non-uniformity of the error bars in the renumbered Fig.5, are multiple:

  • The variability between animals: the four conditions (ND 4h, ND:siAS 4h, ND 24h and ND:siAS 24h) correspond to different animals.
  • The T-DND quantification was done only on a portion of ≈100 mg of each organ (as mentioned in the Materials and Methods), and we cannot exclude inhomogeneous distribution of T-DND within the organ.

We added these comments to Fig.5 caption.

Reviewer 3 Report

The manuscript proposes the use of small nanodiamonds as carriers for siRNA. The major concern of the manuscript is the authors have not demonstrated their hypothesis, that means, the authors proposed very small nanodiamonds expecting their clearance. For that purpose, they should analyse the radiactivity longer than 24 h, as well as no changes were observed between 2 and 24 h. I think this experiment should be prolonged and these data included in a revised version of the manuscript for considering for publication.

Author Response

We thank the reviewer for his remark to which we reply below:

Q: “…the authors have not demonstrated their hypothesis…they should analyse the radiactivity longer than 24 h”

R: The main objective of our study was to demonstrate the in vivo efficacy of a siRNA to inhibit an oncogene target when transported by hydrogenated DND. In this respect, we succeeded.

Such a study needs to be accompanied with the evaluations of elimination and tissue distribution.

The choice of DND as the vector was motivated by the compatibility of their primary size with renal clearance, but the whole study does not fall apart because of the absence of such elimination.

Indeed, a fast-renal clearance would reduce the residency time of the active complex.

The non-detected urinal elimination at 24 h probably favored the inhibition efficacy, while it does not prevent a slow excretion from the organs where DND accumulated, if deagglomeration occurs along the time. In E.K. Chow et al., Sc. Trans. Med. 3, 73ra21 (2011), the authors observed that organ excretion of (intravenously injected) DND takes place on a timescale of about 7 days. Such a slow excretion may be hard to detect in our case due to a faster dynamic of release of labile tritium from the T-DND in the water of urines. Moreover, we observed that DND can be eliminated continuously through the other pathway of the feces, starting from 4h. Therefore, in agreement with the rule of the 3R of animal experiments, we decided not to include groups of animals that would be studied at times longer than the one for which we sought siRNA inhibition efficacy (i.e. 24 h).

In reply to the reviewer 1, we justified in more details in the discussion (lines 591-593) that renal clearance is probably hindered by the small aggregation state of the injected nanoconjugate, based on Rojas et al. [20] observations.

Finally, we thank the reviewer for having pointed out the topic of the elimination time dependence, which is of interest for the feces.

We added to Fig.5b (new biodistribution figure) the content in DND of the accumulated feces during the first 4 h and 24 h. This DND content in the total feces shows a linear evolution with time, in perfect agreement with a constant value of radioactivity in a portion of feces sampled at the rectum (Fig.5a). We have commented this added data in section 3.6.

Reviewer 4 Report

In the manuscript entitled “Delivery of siRNA to Ewing sarcoma tumor xenografted on mice, using hydrogenated detonation nanodiamonds: treatment efficacy and tissue distribution,” the authors investigate the use of nanodiamonds as delivery vehicles for siRNA in vitro and in vivo. In the end, the authors demonstrated that the siRNA delivered via nanodiamonds was able to suppress oncogene expression, the nanodiamond carriers were unable to be excreted in the urine as was hypothesized. The conclusions presented in the manuscript are moderately supported by the experimental results, but would benefit from some additional characterization of the complexed nanodiamond carrier to better examine the size distribution and why the carriers were not excreted via the kidneys. I can only recommend this manuscript for publication in Nanomaterials after the authors address the points outlined below.

  1. The manuscript’s readability could benefit from an additional round of editing to address subject-verb agreement issues and misspellings (i.e. humanly vs. humanely, mussel vs. muscle).
  2. The manuscript referenced items in the supporting information, but the supporting information was not included.
  3. The authors hypothesized that the nanodiamonds would be able be eliminated through the urine based on their 7-nm primary size. The DLS experiments show a size of ~30 nm when using the number distribution as opposed to the more commonly reported intensity or volume distributions. This leads to smaller reported diameters than the intensity and volume distributions. The authors should comment on the appropriateness of this choice given the increased error inherent to using a number distribution due to the potential “loss” of the higher sized peaks when transforming the data from intensity to number distribution.
  4. Lines 341-343. The authors attribute the larger hydrodynamic sizes (30 nm from DLS vs. 7 nm primary size) due to the water molecules attached to the nanodiamonds in solution. Given the ultimate results of the nanodiamonds not able to be excreted via the urine (likely based on size), it would be appropriate for the authors to show the TEM of the nanodiamonds (both H-DND and T-DND) after conjugation to the siRNA to elucidate whether substantial aggregation is occurring.
  5. Figures 4 & 5. Why was only the mass ratio of 5:1 examined? It would have been nice to be able to compare the in vivo data in Figures 4 & 5 to the in vitro data in Figure 3, especially given that the focus of much of the discussion is the in vivo delivery of siRNA with DNDs.

Author Response

We thank the reviewer for giving us the opportunity to provide clearer explanation on the size measurements and on the reason why we chose the mass ratio of 5:1.

Below are our replies to each of his concerns.

Q: “The manuscript’s readability could benefit from an additional round of editing”

R: We have conducted an additional round of editing and corrected the mistakes pointed out.

Q:” the supporting information was not included.”

R: The supporting information (SI) had been attached as an independent file during the submission process. It should have been made accessible from the journal website. For the sake of simplicity, we added these SI at the end of the main text of the revised version, to propose a single file to the reviewer.

Q: Size distribution measurements: intensity vs number.

R: We agree with the reviewer that non-corrected intensity distribution is the most reliable DLS data, since it is extracted from the measurement without hypothesis on the nanoparticle shape. In Fig.2, we already used sizes defined in intensity. The only place where we had used size in number is section 3.1. In the revised version, we replaced the values in numbers (30 and 50 nm) by the ones in intensity (60 and 90 nm) everywhere and we removed the values in number in Table S1 for the sake of clarity.

Furthermore, we doubled-checked the method used by the DLS apparatus software to infer the hydrodynamical diameter from the scattered intensity autocorrelation data. We realized that in Table S1, instead of reporting the cumulant Z-average, we indeed reported the dominant population of a non-negative constrained least squared (NNLS) analysis, which is one of the methods of reference for size-polydisperse suspensions (as discussed in the added Ref. 22). We corrected this mistake in the Materials and Methods. It has no impact on the values of Table S1, which were just not properly defined. To avoid any confusion regarding the size extraction methods, we also decided to remove the PolyDispersity Index (PDI) because it refers only to the cumulant Z-average size, finally not presented.

Q: Lines 341-343. The authors attribute the larger hydrodynamic sizes (30 nm from DLS vs. 7 nm primary size) to the water molecules attached to the nanodiamonds in solution.

R: The 30 nm value (converted in number) from the DLS measurement cannot be only explained by the adsorption of water on DND surface but contribute to it. We do know that we have a mild aggregation of DND by groups of four (in average) primary particles. We added this precision to the discussion (line 509-510).

Q: Given the ultimate results of the nanodiamonds not able to be excreted via the urine (likely based on size), it would be appropriate for the authors to show the TEM of the nanodiamonds (both H-DND and T-DND) after conjugation to the siRNA to elucidate whether substantial aggregation is occurring.

R: It is difficult to disperse nanoparticles on TEM carbon grid where they tend to form aggregates when the solution dries. These aggregates cannot then be distinguished from the one present in the solution.

To get a more realistic measurements of the size of the nanoconjugate aggregates, one can consider TEM observations of their interaction with the cell membrane as we did in our previous study of Ref. 16. The reviewer will be able to see in the Fig.4a of Ref. 16 that H-DND appear as aggregates of size≈50 nm at the cell membrane, which is in agreement with a 30 nm size in solution.

We added this precision at the end of section 3.1.

Q: “Figures 4 & 5. Why was only the mass ratio of 5:1 examined?”

R : The choice of a mass ratio of 5:1 for which free siRNA stay in excess in the suspension during the titration process (Fig.1), is to ensure that all H-DND are covered by siRNA. We believe that such a configuration favors the detachment of siRNA molecules which interact only by a portion of their total length with the H-DND surface. Moreover, the mass ratio of 5:1 also ensures that the suspension injected is not too viscous, since at the given necessary dose of siRNA and volume of solution to be injected, the concentration of H-DND remains low enough to have a small impact on the viscosity of the aqueous suspension. Moreover a low cationic vector concentration has a lower toxic impact on living animals.

Q: “It would have been nice to be able to compare the in vivo data in Figures 4 & 5 to the in vitro data in Figure 3, especially given that the focus of much of the discussion is the in vivo delivery of siRNA with DNDs.”

R: The in vitro data guided us for the choice of the mass ratio, and to check the inhibition efficacy of the siRNA used before extending the experiment to animal. Although in vitro inhibition appears to be similar than the in vivo one, one cannot generally extrapolate in vitro results to in vivo ones, which is why we could comment more. We added this comment at the end of the discussion (lines 615-619) of the revised version.

Round 2

Reviewer 1 Report

The authors carefully replied to the authors concerns. 

Reviewer 3 Report

The authors have revised their manuscript and modified according with reviewers suggestions. In my opinion it is suitable for publication